# First Insight into the Whole Genome Sequencing Whole Variations in *Mycobacterium bovis* from Cattle in Morocco

**DOI:** 10.3390/microorganisms12071316

**Published:** 2024-06-27

**Authors:** Mohammed Khoulane, Siham Fellahi, Slimane Khayi, Mohammed Bouslikhane, Hassan Lakhdissi, Jaouad Berrada

**Affiliations:** 1Department of Veterinary Pathology and Public Health, Institut Agronomique et Vétérinaire Hassan II, Rabat 10112, Morocco; fellahisiham2015@gmail.com (S.F.); bouslikhanemed@yahoo.fr (M.B.); lakhdissi23@gmail.com (H.L.); jaouad.berrada@gmail.com (J.B.); 2Biotechnology Research Unit, Regional Center of Agricultural Research of Rabat, National Institute of Agricultural Research, Rabat 10090, Morocco; slimane.khayi@inra.ma

**Keywords:** *Mycobacterium bovis*, genomics, PCR, WGS, SNP, Morocco

## Abstract

Six cattle heads which tested positive against bovine tuberculosis (bTB) in Morocco were investigated to confirm the disease and to determine the source(s) of infection. Polymerase Chain Reaction (PCR) was directly performed on tissue samples collected from slaughtered animals. All investigated animals tested positive to PCR for the *Mycobacterium bovis* sub-type. Bacteriological isolation was conducted according to the technique recommended by WOAH for the cultivation of the *Mycobacterium tuberculosis* Complex (MBTC). Whole genome sequencing (WGS) was carried out on six mycobacterial isolates and the phylogenic tree was constructed. The six Moroccan isolates fit with clades II, III, IV, V and VII and were confirmed to belong to the clonal complexes Eu2, Unknown 2 and 7 as well as to sublineages La1.7.1, La1.2 and La1.8.2. The significant Single Nucleotide Polymorphism (SNPs) ranged from 84 to 117 between the isolates and the reference *M. bovis* strain and from 17 to 212 between the six isolates. Considering the high resolution of WGS, these results suggests that the source of infection of the bTB could be linked to imported animals as five of the investigated reactor animals were imported a few months prior. WGS can be a useful component to the Moroccan strategy to control bTB.

## 1. Introduction

Bovine tuberculosis (bTB) is a major zoonotic contagious disease of public health concern which affects humans and animals. Besides its public health significance, bTB causes serious economic losses to the cattle industry. The main sources of contamination for humans are the infected animals. The disease affects various livestock species and wild animals.

Bovine tuberculosis is a notifiable, prevalent and enzootic disease in Morocco [1] with potential impact on public health as evidenced by the increase in extrapulmonary tuberculosis cases reported recently [2]. According to a survey carried out in 2003 by ONSSA (National Office of Sanitary Food Safety), the average prevalence of bTB in Morocco was 33% for cattle herds and 18% in cattle population [1,3]. Morocco implemented a strategy to control bTB and adopted several regulations to combat this disease [1]. The strategy is based on the screening of cattle by the tuberculin skin test/gamma-interferon test, and the slaughter reactor animals. Bovine tuberculosis testing of bTB is considered voluntary in Morocco. In slaughterhouses, the search of suspicious bTB lesions is systematic on bovine carcasses. Depending on the nature and extent of bTB compatible lesions, and according to national legislation, affected carcasses are partially or totally condemned.

Numerous cattle farms units in Morocco joined the national program to screen and control bovine tuberculosis [1]. However, this program needs to be reinforced by systematic testing, trace back and determination of the sources of contamination. Epidemiological and analytical investigations (histopathology, PCR and bacterial isolation) to trace infection in newly infected herds are only carried out in rare occasions in Morocco.

Bovine tuberculosis is caused by *Mycobacterium bovis (M. bovis)*, a facultative intracellular parasite and member of the *Mycobacterium tuberculosis* complex (MBTC).

The genome size of *M. bovis* is 4.34 Mpb [4]. Deoxyribonucleic acid (DNA) of members of MBTC have more than 99.9% similarity and contain a high percentage of Guanine and Cytosine (more than 60%). Their genome has a clonal population structure [4,5,6,7,8]. *M. bovis* are classified within different clonal complexes (CCs) and sublineages. The geographic distribution for each CC and sublineage was described [9,10,11,12].

In recent years, several studies have demonstrated the advantages of Whole genome sequencing (WGS) in tracing sources of contamination. WGS has been shown to be more precise than genotyping techniques such as Spoligotyping and Variable Number Tandem Repeat (MIRU-VNTR) [13,14,15,16]. The power and interest of the WGS technique have been demonstrated in the human healthcare sector. Studies have used the WGS technique to identify and trace contaminations in people who have undergone surgical operations. The pathogen studied was *Mycobacterium chimaera* [17,18].

The aim of this study, performed by a Moroccan team, is to use new molecular biology techniques, specifically whole genome sequencing, to characterize the *M. bovis* Moroccan isolates, to identify their phylogenic relationship compared to the strains in public database, and to identify the sources of *M. bovis* infection. This work will improve our knowledge about *M. bovis* in Morocco and promote WGS to strengthen the national strategy to control bTB.

## 2. Materials and Methods

### 2.1. Tissue Samples

The present study conducted between 2014 and 2021 from three (3) regions of Morocco (Casablanca, Gharb and Fes), focused on six (6) cattle heads which tested positive for bTB using intradermal tuberculin and/or gamma-interferon tests as part of the national tuberculosis control program. The six animals, five of which were imported from a European country six months prior, were slaughtered according to national regulations (Table 1). Tissues samples were collected at the slaughterhouses from 6 animals. Respiratory lymph nodes presenting suspicious lesions of bTB (granulomas) were submitted in compliance with national regulations for bacteriological investigations at the Department of Veterinary Pathology and Public Health of the Institut Agronomique et Vétérinaire Hassan II (IAV Hassan II) in Rabat, Morocco.

### 2.2. DNA Extraction

Mycobacterial DNA was extracted directly from tissue for real-time PCR (polymerase chain reaction) detection and from the cultures grown on the culture media as detailed in the Section 2.4 for sequencing investigation; using MagMAX CORE Mechanical Lysis Module Bundle includes using a MagMAX CORE Mechanical Lysis Module and MagMAX CORE Nucleic Acid Purification (Applied Biosystem, Thermofisher, New York, NY, USA) according to the manufacturer’s instructions. The DNA extracted was immediately stored at −20 °C to be used in various molecular methods.

### 2.3. Real-Time PCR Identification Assay

A real-time PCR analysis of MBTC was performed directly from the tissue sample using the VetMAX *M. tuberculosis* kit (Applied Biosytems, Thermofisher, USA). The PCR program consisted of two initial steps of 2 min at 50 °C and 5 min at 95 °C, respectively, followed by 40 amplification cycles (denaturation for 10 s at 95 °C, annealing and extension for 30 s at 60 °C). The reaction was carried out using a Quant Studio 5 real-time PCR system (Thermofisher, USA). Amplifications were recorded and analyzed, and the threshold cycle (Ct) was determined with Quant Studio software (Version 1.5, Thermofisher).

A real-time PCR assay for the differentiation of *M. bovis* from other *Mycobacterium tuberculosis* complex species was carried out using primers and a probe described by Halse et al., 2011 [19]. Briefly, one-step PCR was performed with 12.5 μL of 2× PCR buffer mix, 0.5 μL of MgSO4 (50 mmol/L), 0.5 μL of Rox reference dye (25 mmol/L, Life Technologies, Carlsbad, CA, USA), 0.5 μL of Taq DNA polymerase (Life Technologies), 0.5 μL of primers (0.2 μmol/L), 0.25 μL of probe (0.1 μmol/L), and 5 μL of DNA template to make a final volume of 25 μL. The reaction was carried out using a Quant Studio 5 real-time PCR system (Thermofisher, USA) at 95 °C for 10 min, followed by 45 cycles at 95 °C for 15 s and 60 °C for 1 min. Thermocycling, fluorescence data collection, and data analysis were performed with a QuantStudio 5 system according to the manufacturer’s instructions, with the passive reference dye, ROX, turned on [20,21,22,23]. This is the first time in Morocco that the direct PCR test for animal tissue sample was performed.

In addition, before launching DNA extraction and whole genome sequencing of putative MBTC isolates from bacterial cultures, the cultures were tested by PCR MBTC and the positive ones were tested by the PCR subtype *M. bovis* and subtype *M. tuberculosis*.

### 2.4. Mycobacterial Isolation

Mycobacterial isolation was performed using the technique recommended by the WOAH (alias OIE) for the cultivation of the MBTC described in the manual of diagnostic tests and vaccines for terrestrial animals [24] and the protocol recommended by National Veterinary Service Laboratories (Ames, IA, USA) [25,26]. Two solid culture media were used for growing *Mycobacterium* sp. (Löwenstein Jensen (LJ) and Herold). Proskauer and Beck (P&B) liquid medium was used for subcultures. These culture media were shown in previous studies carried out in the Microbiology laboratory at the Department of Veterinary Pathology and Public Health in IAV Hassan II to be reliable and effective for isolation of *M. bovis* [27,28,29]. Briefly, the collected lymph nodes were processed into a biosafety cabinet class II and surface-decontaminated by using 12° bleach solution diluted to 3% for a few seconds. Lymph nodes were homogenized by grinding into sterile mortar containing a sterile phenol red broth. Homogenized tissue was treated by a solution of 0.5 NaOH for 15 min and neutralized with 6 N HCl. After centrifugation of the suspension at 3000 G for 20 min, the sediments were inoculated into LJ and Herold media. The cultures were incubated for at least 6 weeks at 37 °C and inspected weekly for growth.

Smears from suspected bacterial colonies were stained by the Ziehl–Neelsen method. Acid-alcohol-fast colonies were subsequently subcultured into Herrold, Lowenstein Jensen and Proskauer and Beck (P&B) liquid media to obtain a pure culture yielding a sufficient quality DNA load.

### 2.5. Whole Genome Sequencing

The genomic DNA of the six isolates were extracted using a MagMAX CORE Nucleic Acid purification kit (Applied Biosystems, Thermofisher, Indianapolis, IN, USA) according to the manufacturer’s protocol. Subsequently, a DNA quantity and quality control was checked by measuring the A260/280 ratio using the Nanodrop (ND 8000), quantified fluorometrically using the Qubit 3.0 (Thermofisher, Waltham, MA, USA) and agarose gel electrophoresis at 1%.

The extracted DNAs were then sent to the Eurofins laboratory in Germany for full genome sequencing. The sequencing libraries were prepared with the TruSeq kit (Illumina, San Diego, CA, USA) according to the manufacturer’s protocol. Paired-end sequencing was conducted on an Illumina NovaSeq 6000 system (Illumina Inc.) in 150 bp paired-end mode. 

The raw data of the six Moroccan isolates of *M. bovis* are publicly available in the NCBI Sequence Read Archive under Bioproject PRJNA1099650 and the accessions numbers SAMN40946111 to SAMN40946116.

### 2.6. Genome Assembly, Variants Calling and Profiling of M. bovis Isolates

Prior to the genome assembly, raw reads were trimmed to remove the low-quality sequences using Trimmomatic (version 0.36) with default parameters. The assembly of the trimmed reads was performed using shovill (v0.9.0) available at https://github.com/tseemann/shovill (accessed on 27 October 2023) based on spades software. The generated assemblies were assessed using QUAST (v5.2.0) software with *M. bovis* AF2122/97 as reference [30]. The annotation of six isolates was performed using Prokkav1.14.6 with the following parameters (–genus Mycobacterium–gram pos–usegenus) [31]. Variant calling of the six isolates versus the reference *M. bovis* AF2122/97 was performed using vSNP3 pipeline available at https://github.com/USDA-VS/vSNP3 (accessed on 27 October 2023). The complex clonal and sublineage were performed using TBProfiler software version 5.0.1.

### 2.7. Phylogenomics Analysis of M. bovis Isolates

To obtain insight into the phylogenetic position of the six isolates within *M. bovis*, we retrieved the 46 genome sequences from NCBI including three members of the MBTC *M. tuberculosis*, *M. africanum*, *M. canettii* as an outgroup (Appendix A). The phylogenomics tree was generated based on the shared BUSCO genes within 147 genomes sequences including the six Moroccan isolates. The analysis was performed using the BUSCOPHYLO webserver [32].

## 3. Results

### 3.1. Real-Time PCR Test from Animal Tissue

All animal tissues tested by PCR were positive for MBTC and positive for *Mycobacterium bovis* with Ct values between 23 and 31. The PCR test for *M. tuberculosis* subtype was negative for the six Moroccan isolates.

### 3.2. Sequencing, Assembly and Genome Annotation

The genome assembly of the six isolates was performed using the shovill pipeline after trimming the low quality read sequences. DNA sequencing and assembly generated 4,310,575 to 4,348,552 nucleotides and a GC content between 65.57% and 65.62%. The number of contigs obtained was 82 to 98 and the largest contig ranged from 281,963 to 381,307. From the Quast tool, our *M. bovis* isolates had N50 values between 184,245 and 197,909. *Mycobacterium bovis*_AF2122 was used as a reference isolate. Table 2 summarizes the genome assembly results.

### 3.3. Phylogenetic Analysis

To elucidate the phylogenetic relationship of Moroccan *M. bovis* isolates, we constructed a phylogenic tree based on single-copy BUSCO genes shared across 46 strains from different countries and hosts. A total of 664 BUSCOs were identified as conserved in all genome sequences. By concatenating the alignments into a single sequence per strain, we generate a composite sequence comprising 223,002 amino acids, which served as basis to infer the phylogeny. Within the tree, the six Moroccan isolates were classified into separate clades, revealing three prominent groups. Notably, strains S96K2-20 and S95IM-20 formed a clade within the sublineage La1.7.1, sharing a common ancestor with strains from different countries and hosts, including the more distant Moroccan strains S48B-21 and S68D-21. The strain S69B-21 represented a unique Moroccan lineage within sublineage La1.8.1, harboring several strains from European countries and diverse hosts. Finally, strain S75D-14 stood out as the sole Moroccan representative within sublineage La1.2, clustering alongside strains from European countries and other host backgrounds.

### 3.4. Clades and Clonal Complexes of Moroccan M. bovis Isolates

According to previous studies [13,15] and the SNPs analysis, the six Moroccan isolates are classified into the clades IV, II, III, V and VII. Two isolates (33%) in clade IV, one isolate (16.5%) in clade II, one isolate (16.5%) in clade III, one isolate (16.5%) in clade V and one isolate (16.5%) in clade VII (Table 3). While for the clonal complex, four isolates (66%) are classified at Eu2. The four Moroccan isolates present the mutation G→A on the guaA gene. The TBProfiler software classified the two others isolates. One isolate is classified as a CC unknown 2 and one as a CC unknown 7 (Table 3, Appendix A).

According to the new classification proposed by Zwyer et al. [9], four Moroccan isolates were a part of a clonal complex Eu2 and sublineage La1.7.1. One isolate was a CC unknown 2, sublineage La1.2. The sixth isolate was a CC unknown7, sublineage La1.8.2 (Table 3 and Figure 1).
Figure 1Phylogenetic tree performed by BUSCOPhylo using the maximum likelihood method. The six studied isolates are represented by a gray color. *M. tuberculosis*, *M. africanum* and *M. canettii* are used as an outgroup. The bootstrap values are added. The complex clonal [12,33,34,35,36,37] and sublineage [9,37] for each isolate were added.
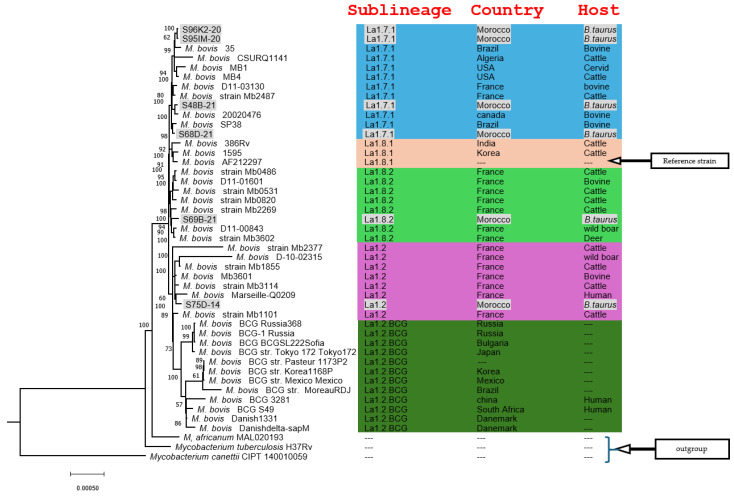

microorganisms-12-01316-t003_Table 3Table 3The clades, clonal complex and sublineage of *M. bovis* Moroccan isolates according to [13,15] and the new classification according to [9]. The CC and sublineage were obtained by the TBProfiler [38].IsolatesClades According to [13,15]Clonal Complex Performed by TBProfiler [38]Sublineage According to [9]S48B-21VEu2La1.7.1S96K2-20IVEu2La1.7.1S95IM-20IVEu2La1.7.1S68D-21IIIEu2La1.7.1S75D-14VIIunknown2La1.2S69B-21IIunknown7La1.8.2


### 3.5. Single Nucleotide Polymorphism Results

Analysis of the Single Nucleotide Polymorphisms (SNPs) of the six isolates sequenced shows a small number of differences (Table 4 and Table 5, Appendix A). The number of SNPs between the six Moroccan isolates and the reference strain *M. bovis* AF2122/97 varies from 276 to 391. The number of SNPs among the compared isolates ranged from 286 to 666 except for the two isolates (S95IM-20 and S96K2-20) where the number of SNPs is only 48.

When focusing on the SNPs of the genes (Table 5), the number varies from 84 to 117 between the six Moroccan isolates and the reference strain. On the other hand, the number of SNPs among studied isolates varies from 17 to 212. The lowest difference of SNPs (17 SNPs) is observed between the isolates S95IM-20 and S96K2-20.

## 4. Discussion

The animals investigated in this study have traceability data including the SNIT number (national identification and traceability system), origin, movements, and age, etc. These epidemiological data allow for conclusions with a high level of accuracy. Bacteriological and PCR investigations confirmed bTB in the six investigated animals. The six isolates obtained from suspicious lesions at the slaughterhouses were confirmed as *Mycobacterium bovis*. The PCR subtype *M. tuberculosis* was negative for the six Moroccan isolates. This suggests that tuberculosis in the cattle subjected to this study has more likely an animal origin rather than human.

The repartition of the six *M. bovis* isolates in several clades suggest multiple origins as the sources of infection of the investigated animals.

Clades are determined in the phylogenic tree by the distinction of broad branches [6]. On the other hand, clonal complexes are groups of bacterial strains descended from a single cell that was the most recent common ancestor of the resulting complex. They share many alleles at several phylogenetically informative loci. A clonal complex includes the ancestral genotype and samples with small variations [35,39,40].

Most studies on *M. bovis* describe four clonal complexes with different geographic distributions and genetic characteristics. The clonal complexes of the *M. bovis* population are African 1 (Af1), African 2 (Af2), Europe 1 (Eu1), and Europe 2 (Eu1) [33,34]. Another clonal complex named African 5 (Af5) was recommended in another study [41].

The Af1 and Af2 clonal complexes are distributed in Africa, whereas Eu1 and Eu2 are distributed in Europe and around the world [12,33,34,35,36].

Distinction between clonal complexes is achieved by determining regions of differentiation on the genome (=RD Regions of Difference) specific to each one. The Eu1 clonal complex has RDEu1 (RD17, 806 bp), Eu2 has a mutation in the guaA gene, Af1 has RDAf1 (5322 bp) and Af2 is characterized by RDAf2 (14,094 bp) [12,33,34,35,36].

Four of the investigated *M. bovis* isolates were found to belong to the Eu2 clonal complex which has been reported in previous studies as being prevalent in Western European countries [36]. The Unkown2 clonal complex was reported in different European countries, South American countries and African countries [9]. Moreover, the Unkown7 clonal complex was reported in different European countries, Tunisia and Mali [9].

The previous studies used a threshold of 12 SNPs to identify strains possibly involved in transmission events [13,42,43]. Another study used 20 SNPs as a threshold to establish a transmission link between *M. bovis* strains [15]

Studies using WGS and SNP analysis were able to establish the relationship of transmission of *M. bovis* through a commercial livestock trade. Perea et al. proved the introduction of a new strain into Spain by animals imported from United Kingdom [13]. Orloski et al. was also able to provide genetic testing of sources of *M. bovis* infection in the USA through cattle imported from Mexico [15].

The analysis of the SNPs of the six Moroccan-investigated *M. bovis* isolates revealed a genetic proximity between the investigated isolates. Two *M. bovis* isolates, S95IM-20 and S96K2-20, originating from two animals have 17 common SNPs which suggest that they probably have the same infectious source. These two animals originating from two different herds were imported on different dates from the same origin. And the two animals were not living together in the same flocks.

All these findings suggest that the sources of infection are linked to cattle importation from European countries.

## 5. Conclusions

This first study illustrates the evolution and the dynamic changes of bTB as well as the genetic diversity of *M. bovis* circulating in Morocco by using WGS. It highlights the usefulness of molecular tools in tracing the sources of infections which can be extended to investigate the sources of zoonotic infection for humans. The inclusion of such tools in official control programs of bTB in animals and humans is to be promoted in Morocco.

## Figures and Tables

**Table 1 microorganisms-12-01316-t001:** Data of the animals studied.

Animal ID	Age at Slaughter	Animal Specie	Sex of Animal	Breed	Farm’s Localization	Origin
S48B-21	<2 years	*Bos taurus*	female	Holstein	North-west (Gharb)	Imported
S96K2-20	2–3 years	*Bos taurus*	female	Holstein	North-west (Gharb)	Imported
S95IM-20	<2 years	*Bos taurus*	female	Holstein	Central (Casablanca)	Imported
S68D-21	3 years	*Bos taurus*	female	Holstein	North central (Fes)	Imported
S75D-14	3 years	*Bos taurus*	female	Holstein	North central (Fes)	Autochthonous
S69B-21	3 years	*Bos taurus*	female	Holstein	North-west (Gharb)	Imported

**Table 2 microorganisms-12-01316-t002:** Assembly and annotation of the six Moroccan isolates. GC (%): percentage of Guanine and Cytosine nucleotides in the reference genome. N50: length for which the collection of all contigs of that length or longer covers at least half an assembly. CDS: coding sequences. rRNA: ribosomal RNA. tRNA: transfer RNA.

Isolate ID	No. of Contigs	Total Length	Largest Contig	GC (%)	N50	CDS	rRNA	tRNA
S48B-21	82	4,340,969	341,813	65.57	197,909	3990	3	52
S96K2-20	82	4,311,726	303,424	65.57	197,698	3964	3	52
S95IM-20	94	4,310,575	302,824	65.58	186,946	3962	3	52
S68D-21	98	4,336,558	281,963	65.60	192,656	4001	4	52
S75D-14	96	4,337,283	341,112	65.62	184,245	4003	3	52
S69B-21	90	4,348,552	381,307	65.58	186,783	3996	3	52

**Table 4 microorganisms-12-01316-t004:** Total SNPs between the *M. bovis* AF2122/97 (reference strain) and the six Moroccan isolates and among the different Moroccan isolates.

	Reference Strain	S68D-21	S95IM-20	S96K2-20	S48B-21	S69B-21
S75D-14	391	659	666	658	657	657
S68D-21	304		347	337	331	568
S95IM-20	309			48	299	576
S96K2-20	301				286	568
S48B-21	302					566

**Table 5 microorganisms-12-01316-t005:** SNPs between the *M. bovis* AF2122/97 (reference strain) and the six Moroccan isolates after filtering and removing the silent mutation and position not annotated.

	Reference Strain	S68D-21	S95IM-20	S96K2-20	S48B-21	S69B-21
S75D-14	117	194	210	208	198	212
S68D-21	84		104	102	92	179
S95IM-20	100			17	97	195
S96K2-20	98				92	193
S48B-21	88					183

## Data Availability

The original data of the six Moroccan isolates presented in the study of the six Moroccan isolates are openly available in [NCBI SRA database] at BioProject accession number PRJNA1099650 (SAMN40946111 to SAMN40946116).

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
