# Peer review of "First Insight into the Whole Genome Sequencing Whole Variations in Mycobacterium bovis from Cattle in Morocco"

_microorganisms, 2024, doi:10.3390/microorganisms12071316_

Round 1

Reviewer 1 Report

Comments and Suggestions for Authors

The manuscript by Mohammed Khoulane et al is devoted to implementation of WGS in tracing the sources of Mycobacterium bovis isolated from infected cows in Morocco. This study is actual in sight of bovine tuberculosis control and prevention. However I have some remarks:

1. The title of manuscript is somewhat ambiguous. One can think it have been first WGS of M. bovis at all, not in Morocco.

2. The Mycobacterial Isolation part, “The sediments were inoculated into LJ and Herold media.” What type of media were used, liquid or agarized? Please clarify.

3. The Phylogenomics analysis of M. bovis isolates part. In “Busco” word all letters must be capital as in Phylogenetic analysis part, because BUSCO is an abbreviation.

Author Response

We are grateful to the reviewer for their valuable comments and for providing us with the opportunity to enhance our manuscript for resubmission to Microorganisms journal. We have thoroughly reviewed the comments and made the necessary revisions to the manuscript. Below, we present a detailed, point-by-point response to all the questions and comments raised by the reviewer.

Comment 1: The title of manuscript is somewhat ambiguous. One can think it have been first WGS of M. bovis at all, not in Morocco.

Response 1: We agree with the reviewer and now we have rectified the title in the manuscript (page 1, lines 1 and 2). It’s read us follow: “First insight into the whole genome sequencing whole variations in Mycobacterium bovis from cattle in Morocco”.

Comment 2: The Mycobacterial Isolation part, “The sediments were inoculated into LJ and Herold media.” What type of media were used, liquid or agarized? Please clarify.

Response 2: The comment of the reviewer is fair, LJ and Herold are solid media. We have now added the specification in the manuscript (page 4, paragraph 2.4, line 132). It’s read as follow: "Two solid culture media were used for growing Mycobacterium sp. (Löwenstein Jensen (LJ) and Herold)".

Comment 3: The Phylogenomics analysis of M. bovis isolates part. In “Busco” word all letters must be capital as in Phylogenetic analysis part, because BUSCO is an abbreviation.

Response 3: Agree with reviewer. we have, accordingly, modified it in the manuscript (page 5; paragraph 2.7, lines 181 and 183).

Reviewer 2 Report

Comments and Suggestions for Authors

The manuscript, submitted by Khoulane and colleagues for publication to Microorganisms, provided an in-depth analysis of the complete genome sequences and phylogenetic analysis of Mycobacterium bovis genomes from six cattle collected in Morocco. They carried out a detailed description of M. bovis genomes molecular epidemiology in this area. The paper is well structured and exhaustive in every part, while the statistical methods are suitable for this kind of investigation. It can be recommended for publication, upon addressing some minors.

-          In the introduction, the authors should also cite other experiences, where phylogenetic analysis was used in molecular epidemiology of emerging Mycobacterium spp. infection (i.e. 10.1016/S1473-3099(17)30324-9, 10.1128/spectrum.02893-22).

-          On page 4, the bioproject PRJNA1099650 has not been released yet. Please, consider opening it as soon. Moreover, in paragraph 2.7, the reference s41598-022-22461-0 should be put into the references section.

-          On page 5 they used an NJ tree, that usually considered as most simple method to build a phylogenetic tree, but have they tried to carry out a Maximum likelihood tree? has the best mutation model been looked at for this dataset? what bootstrap value (number of repetitions) was set for the analysis? 

-          In my opinion, since the MBTC is already partially unknown, this would strengthen the data of phylogenetic analysis.

Comments on the Quality of English Language

None

Author Response

Comments 1: In the introduction, the authors should also cite other experiences, where phylogenetic analysis was used in molecular epidemiology of emerging Mycobacterium spp. infection (i.e. 10.1016/S1473-3099(17)30324-9, 10.1128/spectrum.02893-22).

Response 1:  We thank the reviewer for sharing these references. We have added a paragraph including those articles, and it reads in ((page 2; paragraph 5, lines 69-72) as follows: “The power and interest of the WGS technique have been demonstrated in the human healthcare sector. Studies have used the WGS technique to identify and trace contaminations in person who have undergone surgical operations. The pathogen studied was Mycobacterium chimaera”.

Comments 2: On page 4, the bioproject PRJNA1099650 has not been released yet. Please, consider opening it as soon. Moreover, in paragraph 2.7, the reference s41598-022-22461-0 should be put into the references section.

Response 2: The reviewer is right, we have rectified it. For "opening the bioproject PRJNA1099650" , we modifiy the date of opening of the bioproject PRJNA1099650 in NCBI website, and, the reference s41598-022-22461-0 is put it now into the references section ((pages 5 and 12; paragraph 2.7, lines 183, 437 and 438 )

Comments 3: On page 5 they used an NJ tree that usually considered as most simple method to build a phylogenetic tree, but have they tried to carry out a Maximum likelihood tree? has the best mutation model been looked at for this dataset? what bootstrap value (number of repetitions) was set for the analysis? 

Response 3: Thank you for bringing this to our attention. The reviewer is right, the phylogenetic tree in Figure 1 is built by BUSOPhylo that use the Maximum likelihood method. We corrected the mistake in line 220 (page 7; Figure 1).

Moreover, the bootstrap value are added on the branches of the phylogenic tree. page 8, Figure1.

Comments 4: In my opinion, since the MBTC is already partially unknown, this would strengthen the data of phylogenetic analysis.

Response 4: We agree with Review. This work opens new perspectives to understand the MBTC specially in Morocco.